

# Ground subsidence and heave over permafrost: hourly time series reveal inter-annual, seasonal and shorter-term movement caused by freezing, thawing and water movement

Stephan Gruber[1]

[1]Department of Geography and Environmental Studies, Carleton University, Ottawa, ON, K1S 5B6, Canada
**Correspondence:** Stephan Gruber (stephan.gruber@carleton.ca)

**Abstract.** Heave and subsidence of the ground surface can offer insight into processes of heat and mass transfer in freezing and thawing soils. Additionally, subsidence is an important metric for monitoring and understanding the transformation of permafrost landscapes under climate change. Corresponding ground observations, however, are sparse and episodic. A simple tilt-arm apparatus with logging inclinometer has been developed to measure heave and subsidence of the ground surface with hourly resolution and millimetre-accuracy. This contribution reports data from the first two winters and the first full summer, measured at three sites with contrasting organic, and frost-susceptible soils in warm permafrost. The patterns of surface movement differ significantly between sites and from a prediction based on the Stefan equation and observed ground temperature. The data is rich in features of heave and subsidence that are several days to several weeks long and that may help elucidate processes in the ground. For example, late-winter heave followed by thawing and subsidence, as reported in earlier literature and hypothesised to be caused by infiltration and refreezing of water into permeable frozen ground, has been detected. An early-winter peak in heave, followed by brief subsidence, is discernible in a previous publication but so far has not been interpreted. An effect of precipitation on changes in surface elevation can be inferred with confidence. These results highlight the potential of ground-based observation of subsidence and heave as an enabler of progress in process understanding, modeling and interpretation of remotely sensed data.

## 1 Introduction

The presence, formation and decay of ground ice control many phenomena in the natural and built environment of cold regions. This is especially true at locations with permafrost, where large amounts of ground ice can exist and where its decay may have pronounced effects at the terrain surface (Shumskiy and Vtyurin, 1963; Mackay, 1970; Heginbottom, 1973). In recent decades, the interest in observation (Liu et al., 2010; Bartsch et al., 2019) and numerical prediction (Hwang, 1976; Lee et al., 2014) of ground subsidence is increasing along with the prevalence and magnitude of ground-ice loss due to anthropogenic climate change.

The formation and melt of excess ice, the volume of ice in the ground exceeding the total pore volume under natural unfrozen conditions, is an expression of the redistribution of mass and energy and often a major determinant of surface heave and subsidence in response to freezing and thawing of soil. Excess ice forms through a number of processes including the





growth of needle ice (Outcalt, 1971) and ice lenses (Rempel et al., 2004) and when it melts, the soil consolidates (Nixon et al., 1971). These phenomena are superimposed on the volume changes without redistribution of mass, which are due to thermal expansion of soil materials and the density contrast of water and ice. Together, these processes define the thermo-hydro-mechanical behaviour of the active layer and near-surface permafrost. Here, the existence of an ice-enriched transition zone, alternating between seasonally frozen ground and permafrost on sub-decadal to centennial time scales, has been recognised (Shur et al., 2005). Challenging the simple distinction of active layer and permafrost, the existence of a transition zone points to the importance of its thermo-hydro-mechanical dynamics for linking surface deformation, phenomena of thawing and freezing, and climate change. As a consequence, the analysis and simulation of subsidence due to permafrost thaw may require careful attention not only to the loss of excess ground ice, but also to the dynamics of its episodic growth. While time series of ground temperature are widely available but rare for moisture content, they are nearly nonexistent for surface heave and subsidence.

Most in-situ and remote observations of surface displacement have frequencies ranging from few times per year to multi-annual. A common assumption in the interpretation of such data is that a seasonal signal is superimposed on, and can be distinguished from, the longer-term trend. When only decadal-scale change is of interest and clearly dominates in the observations available, seasonal or higher-frequency signals may be ignored in data interpretation (e.g., O'Neill et al., 2019; Streletskiy et al., 2017). If however, their magnitudes are similar or only short time series are available (c.f., Anonymous, 1969), the shape of the seasonal cycle and its adequate sampling become important during interpretation. The most common assumption is that the seasonal progression of subsidence or heave resembles the square-root of accumulated thawing or freezing degree-days, as predicted by the Stefan equation (e.g., Liu et al., 2010; Bartsch et al., 2019). The validity of this assumption however, is rarely tested and likely differs between locations and years.

By revealing the temporal patterns of ground heave and subsidence, high-resolution data can support improved understanding and modeling of the underlying processes and phenomena. They can complement borehole temperature monitoring with time series of inferred ground-ice loss or aggradation, an important element of the subsurface energy balance and ground thermal regime that is often neglected in observations. Finally, high-resolution data can provide ground truth for other methods of sensing surface subsidence. The potential of high-resolution measurements of surface displacement has been demonstrated for environments with permafrost and with only seasonally frozen soil (e.g., Overduin and Kane, 2006; Harris et al., 2008). These studies measured hourly surface displacement with millimetre-accuracy and revealed sub-seasonal fluctuations as well as differences in seasonal patterns between years. As they require sturdy support frames and elaborate data logging systems, however, such installations are expensive to deploy and maintain.

Better understanding the dynamics of surface displacement in response to freeze-thaw processes superimposed on the potential of net subsidence requires methods that can be applied at many locations and in the long term. This study demonstrates a candidate method for measuring the vertical movement of natural soil surfaces of about $30\,\mathrm{cm} \times 30\,\mathrm{cm}$ with hourly to daily temporal resolution and millimetre-accuracy. Because it is simple, robust and relatively inexpensive, it may be suitable for application at many sites and in the long term, also in association with critical infrastructure. This contribution has three objectives: (1) To describe the tilt-arm method in terms of its design and expected accuracy, and to share lessons learned from its initial deployment. (2) To describe and interpret the features visible in tilt-arm data. As heave and subsidence are driven



by meteorological forcing but also modulated by site conditions, the commonalities and contrasts between sites of varying
similarity will be examined. (3) To test the hypothesis that seasonal vertical ground movement is well approximated by the
Stefan model.

## 2  Background: measuring subsidence and heave

### 2.1  Episodic observation

Methods for observing heave and subsidence at point locations in-situ and areally by remotete sensing are discussed in a recent
review (Arenson et al., 2016). Here, the characteristics of in-situ methods are presented to provide context for the new method
proposed.

Elevation change of the soil surface, or for differing depths beneath the surface when telescoping aluminum tubes are used,
has been measured relative to metal or fibreglass rods anchored in permafrost (e.g., Mackay et al., 1979; Smith, 1985, 1987;

Overduin and Kane, 2006; O'Neill and Burn, 2012). The accuracies reported are $\pm 1 - 5\,\mathrm{mm}$ and the frequency of manual
readings ranges from once every several years to more than ten times per year.

Maximum heave and subsidence, over typically one year, can be recorded with heave sleeves (Nixon et al., 1995) that have
a typical accuracy of $\pm 10\,\mathrm{mm}$ (Nixon and Taylor, 1998). These comprise a small section of metal tubing connected to a grille
resting on the ground surface. As the sleeve moves up and down a reference pipe anchored in permafrost, a protrusion scratches

a painted section on it. Heave sleeves are robust and highly efficient for recording long-term change (O'Neill et al., 2019).

Optical leveling of markers against benchmarks has reported accuracies on the order of $\pm 1 - 20\,\mathrm{mm}$ (e.g., Mackay et al.,
1979; Mackay, 1973). Differential GNSS (Global Navigation Satellite System) surveying has reported accuracies of $\pm 9 -$
$40\,\mathrm{mm}$ (e.g., Little et al., 2003; Lambiel and Delaloye, 2004; Streletskiy et al., 2017).

### 2.2  Automatic observation

Analogue recorders (Matthews, 1967; Fahey, 1974) and later, electronic data loggers with displacement transducers (Matsuoka,
1994; Matsuoka et al., 1997; Hallet, 1998; Matsuoka, 2003; Harris et al., 2007, 2008; Matsumoto et al., 2010) have been used
to measure surface elevation relative to deeply anchored metal frames at hourly intervals. The accuracies reported are on the
order of $\pm 1\,\mathrm{mm}$. In a similar mode of installation, ultrasonic distance sensors, although affected by the presence of snow and
vegetation, have been used to record hourly surface movement with an accuracy of about $\pm 10\,\mathrm{mm}$ (Overduin and Kane, 2006).

In steep unconsolidated materials, subsidence has been revealed from a combination of continuous GNSS and inclinometer
observations (Wirz et al., 2014, 2016a, b) and in bedrock with a combination of multiple crack meters (Hasler et al., 2012;
Weber et al., 2019), where ice loss has also been inferred from ambient seismic vibration (Weber et al., 2018).





## 3 Study area and sites

The sites instrumented are within 25 km from Yellowknife, Northwest Territories, Canada. Climate is continental subarctic
with a mean annual air temperature of -3.6 °C and an average annual precipitation sum of 291 mm observed at the airport
(YZF) during 1971–2000. The maximum mean monthly snow depth of 0.39 m is usually reached in February. Permafrost near
Yellowknife exists mostly in black spruce forests underlain by fine-grained frost-susceptible soils, and in open black spruce
forest peatlands (Morse et al., 2016). Active layer thicknesses were 0.6–1.2 m at mineral soil sites and 0.5–0.7 m in peatlands.

Three sites have been instrumented (Table 1) on 6–8 July, 2017. *Drill the Chill* is in a peatland with open black spruce and
tamarack forest, 8 km west of Yellowknife. The organic layer has a thickness of about 1.1 m and is underlain by reworked
glacio-lacustrine silt; the active layer is about 0.7 m thick. *Evil Peat* is in a peatland with open black spruce and tamarack
forest, 21 km northeast of Yellowknife. The organic layer has a thickness of about 1.5 m, followed by an interval of fluvial
sand with a thickness of 0.35 m and glacio-lacustrine silt below. The active layer is about 1.2 m thick. *Active Slayer* is located
in dense black-spruce forest 17 km northeast of Yellowknife, with about 0.1 m of organics overlying hummocks in mineral
soil. The active layer is about 1.3 m thick beneath hummock tops.

**Table 1.** Sites details, including coordinates (west, north), serial numbers of instruments, active-layer thickness (ALT) and the approximate
thickness of the organic layer overlying mineral soil.

| Site name | Coordinates | Serial | ALT | Organic |
|---|---|---|---|---|
| Evil Peat | 114.016, 62.554 | 5120 | 1.2 m | 1.5 m |
| Drill the Chill | 114.532, 62.457 | 5121 | 0.7 m | 1.1 m |
| Active Slayer | 114.095, 62.540 | 5122 | 1.3 m | 0.1 m |

## 4 Materials and methods

### 4.1 Tilt arm

The vertical movement of the soil surface is measured via the changing inclination of a tilting arm that connects a reference
point with a small grille buried just beneath the vegetation mat at a distance of 1.5 m (Fig. 1). The tilt arm was constructed
from hard wood (aged red oak) to have minimal thermal expansion, and coated with urethane varnish for waterproofing. The
reference point is given by a pivot attached to pipe anchored in permafrost at depth. When tilting, the changing horizontal
distance between the anchor pipe and the ground attachment is accommodated by a double-pivot mechanism. This design
minimizes the disturbance of the surface and subsurface at the point measured. It has few moving parts, is simple to build and
suitable for anchoring on borehole casing pipes used for thermal monitoring or as benchmarks. The red dots visible on top of
the anchor point (Fig. 1B) and the soil attachment (Fig. 1C) facilitate surveying or optical leveling to relate the installation to
external benchmarks. This may help to detect if the anchor pole was frost jacked.



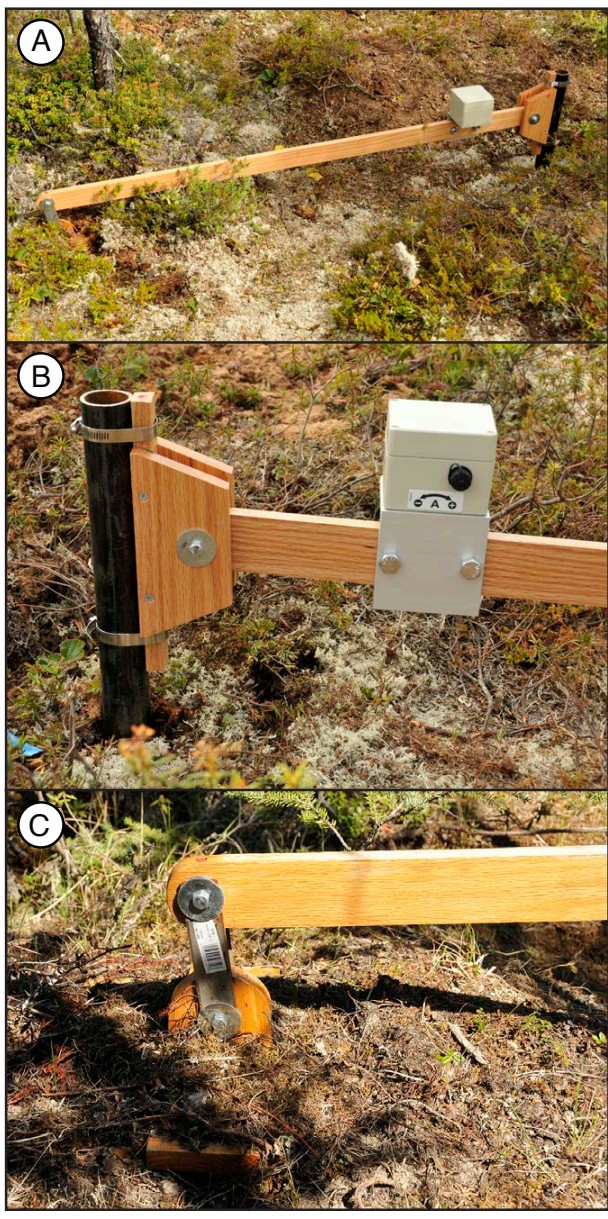

**Figure 1.** Tilt arm installed at site Evil Peat (A) with detailed views on the logger and the attachment to the anchor pipe (B) and the double-pivot mechanism (C) just above the wooden grille inserted beneath the vegetation.

A metal pipe with a length of about three metres was installed at each site to provide anchoring in permafrost at depth. For this, holes with an approximate diameter of 10 cm were drilled using a drill similar to the one described by Calmels et al. (2005). After inserting the pipes, the holes were filled with coarse sand to avoid heaving phenomena during refreezing.





A tilt logger, RST Model IC6560, with a Micro Electrical Mechanical System (MEMS) accelerometer and $4\,\mathrm{MB}$ of memory is used to record hourly time series of inclination. It measures one axis with a range of $\pm\,15\,^\circ$ and has an operating range of -40 °C to 60 °C. The instrument has a resolution of $\pm\,0.0006\,^\circ$ and a repeatability of $\pm\,0.002\,^\circ$ as stated by the manufacturer. When applied to a near-horizontal tilt arm of $1.5\,\mathrm{m}$ this is equivalent to resolving elevation differences of $\pm\,0.015\,\mathrm{mm}$ with a repeatability of $\pm\,0.045\,\mathrm{mm}$. When mounted on the tilt arm, positive inclination corresponds to lowering of the end embedded

in the soil.

## 4.2   Auxiliary data

At a distance of $10\,\mathrm{m}$ or less, soil temperature has been logged 0.30–0.35 m deep. The digital sensors are accurate to $\pm 0.1\,^\circ\mathrm{C}$ and measure hourly. Daily total precipitation and snow height are available for the Yellowknife airport (Government of Canada, 2019).

## 4.3   Conversion from inclination to elevation difference

Measured inclination is converted to elevation change based on the geometry of the tilt arm (Fig. 2) with $L = 1.5\,\mathrm{m}$ and $P = 0.115\,\mathrm{m}$. The vertical distance of the arm's end from the reference point is $dY_L = \sin(\alpha) \times L$, based on the angle $\alpha$ [°] between the horizontal plane and the inclined arm with length $L$. The changing horizontal distance during tilting is $dX = (1-\cos(\alpha))\times L$. The vertical distance between the arm's end and the ground attachment, added by the double-pivot mechanisms

with length $P$, is $dY_P = \sqrt{P^2 - dX^2}$ using the Pythagorean theorem. The vertical distance between the reference point and the ground attachment can then be calculated as $dY = dY_L + dY_P$, or

$$dY = \sin(\alpha)\,L + \sqrt{P^2 - (1 - \cos(\alpha))^2 L^2}\,. \tag{1}$$

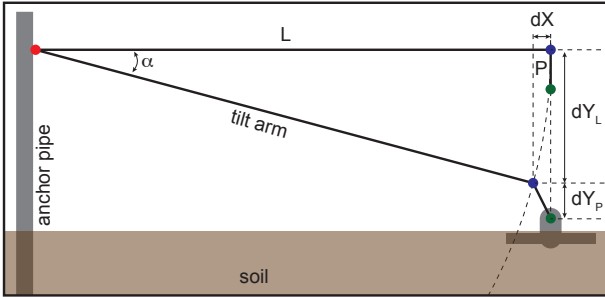

**Figure 2.** Geometry of the tilt arm defined by the lengths of the arm ($L$) and the double-pivot mechanism ($P$). The reference point on the anchor pipe is shown in red, the end of the tilt arm in blue and the ground attachment at the end of the double-pivot in green. Only vertical soil movement is measured.





## 4.4 Error from thermal expansion and contraction

The vertical error introduced by thermal expansion of the anchor pipe is $\varepsilon_{AP} = dT\,\beta\,A$, with $dT$ [°C] being the temperature
difference causing expansion, $\beta$ [m/(m K)] the coefficient of linear thermal expansion and $A$ [m] the length of the anchor pipe
between the permafrost and the elevation of the tilt arm reference point. As permafrost becomes near-isothermal and soft, the
actual anchoring point may be deeper than the permafrost table. While this may be a problem for the anchoring of the pipe,
temperature fluctuations will be subdued. To generate an extreme (high) estimate of $\varepsilon_{AP}$, let the active layer be 1.2 m thick
and entirely at the shallow soil temperature measured, and the length of anchor pipe exposed above the ground surface be 0.5
m and at the temperature measured inside the tilt logger. The temperature-related vertical error introduced by the tilt arm is
approximated as $\varepsilon_{TA} = dY - dY(T)$ based on Equation 1, where $dY(T)$ denotes the temperature-affected vertical distance
obtained from using the thermally expanded lenghts for the tilt arm $L(T)$ and double-pivot $P(T)$. As $\varepsilon_{TA} \propto \alpha$, only values
for $\alpha = 15°$ are reported; the tilt arm is assumed to be at the temperature measured by the logger. The maximum ranges of
instantaneous temperatures in soil (41 °C) and in the logger (85 °C) were measured at Active Slayer, and for these, estimated
errors for typical materials are listed in Table 2. The assumptions on temperature fluctuation and angles are extreme and
represent a deliberately high estimate of error induced by thermal expansion of the measurement setup. Even so, the combined
error from tilt arm and anchor pipe is ±0.6 mm and, therefore, neglected in the results presented.

**Table 2.** Estimated maximum error resulting from thermal expansion of anchor pipe $\varepsilon_{AP}$ [mm] and tilt arm $\varepsilon_{TA}$ [mm] for common materials,
together with typical values of the coefficient of linear thermal expansion $\beta$ [$10^{-6}$ m/(m K)]. Materials used in this study shown in bold
font, note that the short double-pivot pivot segment is made from steel.

| Material | $\beta$ | $\varepsilon_{AP}$ | $\varepsilon_{TA}$ |
|---|---|---|---|
| Plastic (ABS, PVC) | 80 | ±3.74 mm | ±1.69 mm |
| Aluminium | 22 | ±1.03 mm | ±0.46 mm |
| Steel | 12 | ±**0.56** mm | ±0.25 mm |
| Wood, along grain | 3 | ±0.14 mm | ±**0.06** mm |

## 4.5 Error from temperature stability of tilt meter

Laboratory testing was performed to investigate temperature-related artifacts in the inclination recorded. Instruments were
tested in a Caron 7900-25 freeze-thaw chamber at an inclination of 0° while changing temperature from 20 °C to 0 °C to
-20 °C with three hours at each temperature and two hours for transitions. Temperature was monitored using GeoPrecision M-
Log5W-SIMPLE-US miniature data loggers with a precision of 0.01 °C and an accuracy of ±0.1 °C. Inclination is converted
to an equivalent elevation difference $dY = sin(\alpha) \times 1.5$ m for easy comparison with the field installation 3. The stability
of inclination between -20 °C and 20 °C, after few hours of equilibration, was within ±0.8 mm. In the presence of strong
temperature gradients ($\approx 10$ °C h$^{-1}$), differences up to ±2.5 mm were observed. The four tilt loggers tested showed no uniform





response in their error. By comparison, the manufacturer states a repeatability equivalent to $dY = 0.045\,\mathrm{mm}$, whether this includes the effect of variable temperature and temperature gradients is unknown. As the laboratory testing is not based on an inclination reference with known accuracy, there is a possibility that the higher temperature-related error reported here is an artifact of the freeze-thaw chamber deforming during temperature cycles. For the current work, these tests at least highlight the importance of temporal temperature gradients in affecting tilt measurements and they provide a conservative (high) estimate of the error introduced.

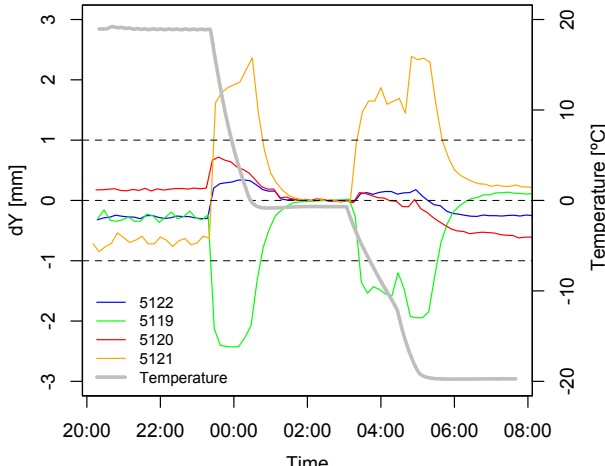

**Figure 3.** Tilt measured by differing sensors in the laboratory. The instruments, identified by their serial numbers, have not been moved during the experiment; instrument 5119 is not used in the field. Inclination is expressed as equivalent elevation change $dY$ assuming an arm of length $1.5\,\mathrm{m}$ as is used in the field.

## 4.6 Data processing

Hourly data are read from the original logger files and converted to elevation differences relative to September 15, 2017. Hourly soil temperatures, recorded by a separate instrument, are interpolated linearly to the times logged by the tilt meters at each location. Hourly data is aggregated to daily values using the arithmetic mean of soil temperature. For elevation differences the median is used as it is more tolerant to extreme values caused by fast heating and cooling. Diurnal ranges of surface elevation and logger temperature are derived based on the daily minimum and maximum values after de-trending the original time series with a centered 25-hour running mean.

## 4.7 Estimating vertical movement from surface temperature time series based on the Stefan equation

Under the assumptions of saturated conditions, negligible amounts of liquid water in frozen soil, and no water movement, the progressive top-down thawing of soil can be related to an equivalent subsidence $s(t) = z(t)\phi(1 - (\rho_i/\rho_w))$ where $t$ denotes





time, $z(t)$ is the thickness of soil thawed during the time interval $[t = 0, t]$, $\phi$ is the porosity of the soil and $\rho_i$ and $\rho_w$ are the densities of ice and water.

The Stefan equation relates the evolution of surface temperature to the progression of the freezing or thawing front in the
soil. When the absolute depth of freezing or thawing is required, terms accounting for the thermal conductivity of the soil between the surface and the frozen/unfrozen interface and for the volumetric latent heat of the soil undergoing phase change as well as correction factors to account for the heat capacity of soil exist (Kurylyk and Hayashi, 2016). Here, only the shape of the temporal progression of subsidence or heave $z = \sqrt{I(t)}$ is used, where $I(t)$ is the thawing index as derived from the cumulative summation of daily positive temperatures or the freezing index based on negative temperatures.

Freezing indices $[°C\,d]$ for each winter were derived from the sum of negative daily temperatures during nine months after September 1 and thawing indices for summer from positive daily temperatures in nine months after March 1. The start and end days of the freezing and thawing seasons are found by using minimum and maximum of each series, excluding values of zero. For display as time series, indices are scaled to coincide with the observed surface elevation on those dates.

### 4.8 Interpreting temperature and vertical movement

The success of predicting observed vertical movement with the Stefan equation and observed near-surface ground temperature is assessed visually. Additionally, prominent features in the observations are identified and hypotheses on their drivers and determinants are developed based on known processes and phenomena.

## 5 Results and interpretation

### 5.1 Diurnal movement

The diurnal range of movement is almost always less than $1\,mm$ (Fig. 4) and the diurnal temperature range in the tilt loggers is mostly less than $30\,°C$. A significant fraction of the diurnal oscillation is likely an artifact, related to thermally-conditioned changes of installation geometry and measurement electronics. This is in line with the error estimates presented in Sections 4.4 and 4.5 and as a consequence, diurnal patterns are not interpreted here.

### 5.2 Detailed time series

The detailed movement and relevant auxiliary information are displayed in (Fig. 5). Here, the prominent features identified in the figure are described and for some, hypotheses on their drivers and determinants formulated. Approximate near-surface zero-curtain periods, the duration of ground freezing and snowpack ripening are inferred visually.

At Active Slayer, a short period of increasing heave is offset partially by subsidence (A) before relative stability for the rest of the first winter. In the second winter, there is a similar initial period of heave (B), followed by long stability. In contrast to
the previous year, there is no subsidence after the initial heave and the overall amount of heave is about three times larger. The start of both periods of stability is synchronous with the establishment of a thick snow cover (Yellowknife airport) although this





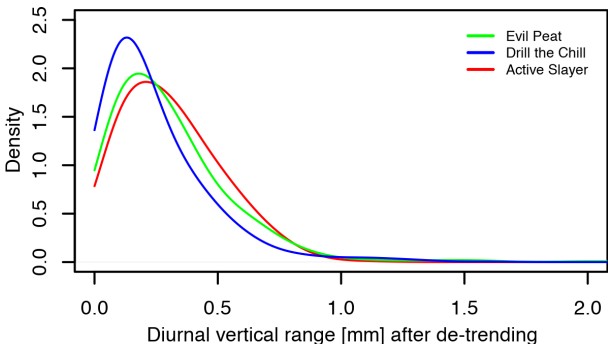

**Figure 4.** Density of diurnal vertical movement for all three sites.

may be coincidental. During summer, subsidence resembles the shape expected based on the Stefan equation but is interrupted by a period of stability (C) and a similar effect may be visible near the end of the time series. This effect may be related to the vertical distribution of ice in the soil, with a desiccated zone, which does not consolidate upon thaw, at intermediate depth in

the active layer.

At Drill the Chill, a period of heave exists after snowpack ripening during the first winter (D). It is followed by the onset of subsidence, while near-surface ground temperature remains isothermal. This pattern may derive from the refreezing of snowmelt infiltrating into frozen soil and subsequent progressive thaw that has not yet reached the depth of the temperature sensor. An initial phase of heave (E), followed by minor subsidence is prominent in the first winter and detectable in the second.

While reminiscent of feature (A), all three instances differ in shape, magnitude and duration. Toward the end of the second winter, a period of relatively steady heave and a total magnitude of about $15\,\mathrm{mm}$ (F) exists. While features (D) and (F) may be interpreted as phase change near the bottom of the active layer, where saturated conditions are frequently found and freezing would commence only late in winter, the relatively rapid rate of heave contrasts with the slow progression of freezing and heave to be expected late in winter based on the Stefan equation.

Sudden heave followed by gradual subsidence is identified by (G) and these features are abundant both at Drill the Chill and at Evil Peat. For a subset of features, thin blue dotted lines highlight the synchronicity of heave between sites and with heavy precipitation recorded at the Yellowknife airport. This synchronicity and the resemblance of the subsidence phase and recession curves support an attribution to hydrologic drivers. As such, also less obvious features observed need to be considered as possibly related to the water balance, including lateral flow. This may offer an explanation for the anomalous thaw season at

Evil Peat, where net heave instead of subsidence was observed in summer 2018 (H) in contrast to the partial summer of 2017 (I). Finally, Drill the Chill and Evil Peat have a synchronous episode resembling a dip (J). Here, subsidence may coincide with the onset of thaw and heave with the infiltration and refreezing of early spring precipitation.





**Figure 5.** Daily elevation change and ground temperature (0.30–0.35 mm) recorded at the three sites, with nearby (<25 km) precipitation (blue steps) and snow height (grey polygons). Air temperature (black) is shown relative to 0°C (dashed) for context and without scale. Blue shading indicates approximate period of soil freezing, orange vertical lines indicate the approximate timing of snowpack ripening and meltwater input into the ground, vertical dashed lines indicate one full measurement year (black), selected precipitation events (blue) and the approximate duration of the autumn zero-curtain periods at Drill the Chill and Evil Peat (red). Letters in circles identify events discussed in the main text.



### 5.3 Comparison with Stefan equation

The observed elevation change deviates from estimates using the Stefan equation. Figure 6 shows observed elevation change

and scaled predictions, shifted to coincide with observations at the start and end of the freezing and thawing seasons. First, the general seasonal evolution of observation and model differ in most cases and in one season, prediction and observation have opposite sign. Second, some observed features of heave and subsidence last several days to several weeks and cannot correspond to the monotonous progression described by the Stefan equation. It is therefore likely, and illustrated well by Figure 5, that processes other than volume changes due to the advance of freezing and thawing have additional, and possibly

dominating, influence on the observed patterns.

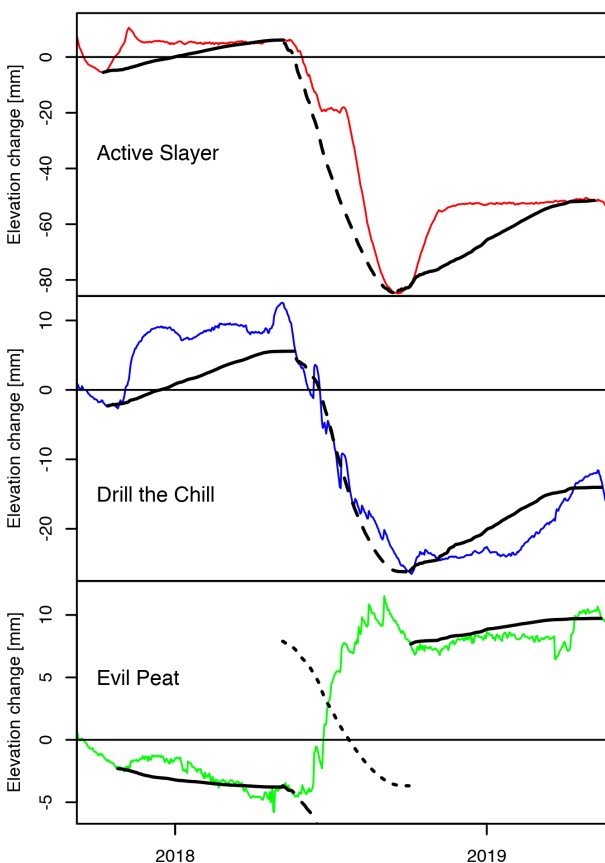

**Figure 6.** Visual comparison of observed elevation change (colour) with predictions (black) based on the Stefan equation and ground temperature. Solid black lines represent freezing seasons and dashed lines thawing seasons. The dotted line for site Evil Peat has been shifted upwards for clarity, illustrating that the observed thawing season was accompanied by surface heave, whereas the Stefan model predicts subsidence. Predictions are scaled and shifted to coincide with observations at the start and end of the freezing and thawing seasons and, therefore, can only be used to visually assess how well they resemble the temporal pattern of elevation, but not seasonal magnitude.



## 6 Discussion

### 6.1 Findings in relation with previous work

The changes in surface elevation between both winters range from 10 mm heave to 60 mm subsidence, similar to inter-annual changes reported from other studies based on continuous (Overduin and Kane, 2006) and annual (e.g., O'Neill et al., 2019) ob-
servations in permafrost areas. Similarly, the seasonal amplitudes of 15–60 mm are within the range of 20–120 mm commonly reported for other, sometimes vegetation free and seasonally-frozen areas (e.g., Matsuoka, 1994; Hallet, 1998). By contrast, the diurnal range of heave reported previously is much larger than the present study, for example 10–40 mm (Matsuoka et al., 1997). This is likely because those studies focused on frost heave in bare soil, which has temperature cycles that are much less dampened than at the sites reported here.

An early-winter peak in heave followed by brief subsidence during early-winter, similar to feature (A) in Fig. 5, was previously detected at a seasonally-frozen and near-horizontal site (Matsuoka, 1994) but not interpreted. At other seasonally frozen sites (Matsuoka et al., 1997; Matsuoka, 2003; Matsumoto et al., 2010) and one in permafrost (Hallet, 1998) however, winters showed gradual early heave, without an early-winter peak. These previously studied sites had frost-susceptible fines like Active Slayer, but negligible vegetation or organic cover.

Late-winter heave followed by thawing and subsidence similar to (D) and possibly (F) in Fig. 5 has been reported previously (e.g., Matsumoto et al., 2010) and, because of its association with the zero-curtain period, also hypothesised to be caused by infiltration and refreezing of water in permeable frozen ground (Matsuoka et al., 1997; Matsuoka, 2003) or by ice segregation at shallow depths following initiation of surface thaw (Harris et al., 2008).

   A late-winter dip, similar to feature (J) at Drill the Chill and Evil Peat, was observed previously at sloping and seasonally-
frozen sites Matsuoka (1994); Harris et al. (2008) but was not interpreted. These sites had frost susceptible fines, whereas the observations of (J) were made on organic material at least 1 m thick. Hydrologic control on elevation change, as hypothesised for (G) and (H), has been described for peat on scales from hours to years (c.f., Price and Schlotzhauer, 1999).

### 6.2 Differing environments and applications

The vertical movement of the soil surface is, to various degrees, determined by climate, the characteristics and stratigraphy
of ground materials, hydrology, and possibly vegetation. The study presented here samples three sites in warm, and likely thawing, permafrost. Many of the interesting geocryological phenomena, including summer heave (Mackay, 1983), the growth of segregated near-surface ground ice (O'Neill and Burn, 2012), the formation of injection ice (Morse and Burn, 2014) or differential heave and settlement in patterned ground, including experimental site manipulation (Kokelj et al., 2007), may be studied with tilt arms at other locations. Similarly, phenomena of seasonal freezing (e.g., Harris et al., 2008) may be
investigated.

   As permafrost areas progressively enter a state of pervasive thaw and ground-ice loss, quantifying and understanding surface subsidence and heave becomes increasingly important. While episodic measurements may reveal multi-annual trends, it will likely be high-resolution observations that can drive progress in simulation of subsidence and the interpretation of remote





sensing data. This method and the results of this study may be relevant for other applications such as correcting snow-height
observations for the heave and subsidence beneath the snow pack, monitoring peatland hydrology and seasonal frost heave in
non-permafrost areas, or providing ground-truth for remote sensing studies. The trend of decreasing cost for tilt sensors and
data logging systems is expected to continue and will make the wide replication of this method increasingly affordable.

### 6.3 Improving the tilt-arm setup

Wood has been chosen as the material for the tilt arm due to its low coefficient of thermal expansion. However, deterioration
of the varnish and ingress of moisture over time may lead to longitudinal expansion of up to 0.4% (Hann, 1969), equivalent to
an additional elevation error of $\pm 0.2\,\mathrm{mm}$ and also, an unknown amount of warping may occur. As a consequence, steel pipe is
the preferable material for future tilt arms.

The anchor pipes in this experiment are relatively shallow and long-term installations will benefit from deep anchoring
because the possibility of upward vertical displacement (frost jacking) has to be taken into account. This is because anchor
pipes embedded insufficiently deep in permafrost may not have the resistance to counteract heaving forces developed in the
active layer (c.f., Ladanyi and Foriero, 1998). The establishment of additional nearby benchmarks, in bedrock or with other
pipes, is useful to support the regular surveying of the tilt-arm installation. Periodic surveying will allow to ascertain the
integrity of long time series by demonstrating no frost jacking or lowering of the pipe due to thaw have occurred. Given
the absence of experience with miniaturized tilt loggers operating in cold environments in the long term, measuring tilt-arm
inclination with a manual inclinometer or digital level during field visits is advisable as it will permit to detect if significant
sensor drift should occur over time.

Lateral movement of the soil can occur on sloping surfaces (e.g., Harris et al., 2007) or be caused by lateral heterogeneity
(e.g., Kokelj et al., 2007; Kääb et al., 2014). Where lateral movement occurs, tilt-arm installations may need to be serviced
episodically to maintain a favorable geometry. Where lateral heterogeneity of materials and ground conditions is large, e.g.
in hummocky terrain or in forest, multiple arms may be needed to adequately represent one terrain type and additionally, the
lateral tilt of shallow stakes may reveal growth or collapse of hummocks. Similarly, an instrument resembling a heavemeter
(e.g., Mackay et al., 1979) can be composed of multiple tilt arms to measure differential frost heave. Finally, given that an
influence of water-table fluctuations was inferred, the direct measurement of water level at tilt-arm installations may prove to
be valuable.

### 7 Conclusions

This study demonstrates a simple method for obtaining high-resolution time series of surface elevation change and, based on
analysis of the data obtained, supports these conclusions:

1. The tilt-arm setup can efficiently monitor surface heave and subsidence over time. The accuracy of daily median values,
   conservatively expressed as the maximum error expected from the effects of thermal expansion and sensor stability over



the entire measurement period, is $\pm 1.4\,\mathrm{mm}$. Hourly values are subject to additional variation of $\pm 0.5\,\mathrm{mm}$ that is largely an artifact from strong thermal gradients.

2. The observed seasonal patterns differ strongly from the prediction of a simple Stefan model and observed near-surface ground temperature. Consequentially, processes other than surface-driven phase change in homogeneous soil are important for explaining surface movement.

3. High-resolution surface heave and subsidence data reveal features that may help to elucidate underlying processes or to discover similarities in space and time. Some of these features have been previously observed but not interpreted in detail.

4. Inter-annual, seasonal and sub-seasonal patterns of subsidence and heave differ in magnitude and sign between the three sites, despite their proximity. This underscores the need to better understand the freeze/thaw processes driving this spatial 305 and temporal heterogeneity. Furthermore, it points to the importance of remote sensing for investigating spatial patterns of terrain subsidence and heave.

5. The interpretation of surface movement as a proxy for freezing and thawing processes can be confounded by hydrologic effects.

*Code and data availability.* Field data, including R-code to generate the plots, is available from https://dx.doi.org/10.5281/zenodo.3466097.

*Competing interests.* The author is affiliated with Cryogeeks Ltd, a company that sells the Geoprecision products mentioned in this article and develops competing products for the tiltloggers described here.

*Acknowledgements.* Thomas Knecht performed the laboratory tests. During the field campaign, Emilie Stewart-Jones, Ariane Castagner, Thomas Knecht and Stuart MacDonald helped with drilling and installation of instruments, Rupesh Subedi, Christian Peart and Nick Brown installed some of the temperature monitoring sites used here. And who came up with those site names? Equipment was available via the 315 project 'Quantifying the Hidden Thaw' financed by the Canada Foundation for Innovation and the Ontario Research Fund. Further support available from the Natural Sciences and Engineering Research Council of Canada. The Northwest Territories Geological Survey supported field logistics and Steve Kokelj gave advice on site selection. Bill Leard built the tilt arms and insisted rightfully that they must have the double-pivot. Thank you to Bin Cao, Jayson Eppler and Steve Kokelj for their comments on the manuscript.





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
