# Peer review of "Ground subsidence and heave over permafrost: hourly time series reveal inter-annual, seasonal and shorter-term movement caused by freezing, thawing and water movement"

_The Cryosphere, 2019_

## Referee Comment (RC1) · Anonymous Referee #1 · 29 Dec 2019

The manuscript by Gruber describes a new device developed to measure heave and subsidence with high resolution in environments dominated by freeze-thaw processes. The device is described and tested in three contrasting sites, and the results are compared to a simple empirical model. The vertical movement of the ground is an important process in these environments, with geomorphological and geotechnical significance.

The manuscript is mainly well written and structured, and comes to relevant results, which may be of interest for the scientific community working with frost heave and also interpreting outcomes from remotely-sensed imagery. The results are discussed

upon relevant other literature, but maybe a bit simplistic against the Stephan solution etc. My impression is somehow positive, however, the authors may think about the following points:

- As always when combining new development of technical devices with scientific results, the resulting papers may lack focus. In this case, the description and development of the tilt arm is combined with first tests and results, and the paper focus on the latter in the end. A part of the discussion is again used to discuss the design of the tilt arm. The introduction of the new device would justify a short communication or similar, while the 2 years data of soil heave may not justify a publication in a high-ranked international journal, even the data look reasonable, and Fig. 5 is illustrative. The results obtained are interesting, and discussed against the possibility of hydrological processes being a major driver. However, this aspect is not documented by measurements or models.

- Introduction: Wordy and could by straightened up, and be merged with "Background" chapter. I miss a key map, and an image over the study area to get an impression how it looks like there.

- Material and Methods: Now, the tilt arm is explained and introduced as a new field device, and the error range is presented well. See comment above. L. 113 mentions a drill and a references, better to explain the drill if it is important.

- Delete the 4.2. paragraph (just one sentence) or incorporate elsewhere. 4.8. the same, paragraph is not necessary.

- Fig. 4: What is "density" here?

- Acknowledgements: l. 315: "And who came up with those site names..". What is this?

---

## Referee Comment (RC2) · Anonymous Referee #2 · 18 Mar 2020

The manuscript by Gruber presents high-resolution measurements of ground subsidence and heave over permafrost soils at three different sites. A new measurement device has been developed and tested in order to measure these soil vertical movements.

Overall, the manuscript is well written and the flow is structured and coherent. The tilt-arm device is well presented, and its potential limitations as well. The accompanying data of heave and ground subsidence are well explained. I encourage the author to make a short communication exclusively on this topic.

[Figure]

One of the main unclear points of the manuscript is Figure 4 and the accompanying text, that must be rewritten. In particular, what does 'density' refer to ?

Some other details need to be addressed by the author, such as :

- Section 3 : Is it possible to add a picture and/or a satellite image of the studied sites ?

- Section 4.2 (auxiliary data) : Splitting soil temperature data section into a whole paragraph does not seem necessary. Could it be merged into section 4.1 ?

- Section 4.8 : paragraph not necessary. Could be removed, or used as a small introductory sentence for section 5

- Section 5.3 : How are defined the thawing and freezing seasons ? Soil temperatures, air temperatures, freezing indexes ?

Moreover, some mathematical notations are unclear, and a few sentences need clarification. For instance :

- Line122 : 'at a distance of 10m or less' : I guess it is at a distance of the tilt-arm logger ? Please precise.

-Line 172 : The wording 'during the time interval $[t=0,t]$' is unclear (and mathematically awkward) : does $z(t)$ simply denotes the thickness of soil thawed at the time $t$ ? If so, please remove the sentence 'during the time interval $[t=0,t]$'.

- Line 178 : Should not it be 'proportional' instead of 'equal' in $z = \sqrt{I(t)}$ ?

- Line 195 : extra parenthesis in the text (Fig.5)

Please check the annotations and remarks in the attached document.

Please also note the supplement to this comment:
https://www.the-cryosphere-discuss.net/tc-2019-227/tc-2019-227-RC2-

supplement.pdf

[Figure]

**Supplement:**

[revised manuscript text omitted]

---

## Author Comment (AC1) · 19 Mar 2020

Thank you very much for your review and your constructive comments on this manuscript. I hope that the explanation given below, and the changes to the manuscript, will provide an adequate response.

Referee comments are indicated as "RC" and author responses as "AR".

RC: As always when combining new development of technical devices with scientific results, the resulting papers may lack focus. In this case, the description and development of the tilt arm is combined with first tests and results, and the paper focus on the latter in the end. A part of the discussion is again used to discuss the design of the tilt arm. The introduction of the new device would justify a short communication or similar, while the 2 years data of soil heave may not justify a publication in a high-ranked international journal, even the data look reasonable, and Fig. 5 is illustrative. The results obtained are interesting, and discussed against the possibility of hydro-logical processes being a major driver. However, this aspect is not documented by measurements or models.

AR: Both referees allude to the possibility of making this a short communication and I respectfully disagree with this suggestion: First, the journal guidelines for (2–4 pages, up to 3 figures/tables, up to 20 references) would not permit an adequate level detail. Second, the potential of processes other than phase change in soil to obfuscate analy-ses of surface movement can be clearly shown, even with this short dataset and in the absence of additional observations of hydrology.

RC: Introduction: Wordy and could by straightened up, and be merged with "Back-ground" chapter.

AR: The introduction has been shortened although it still needs some space to outline the problem to be addressed in the context of relevant previous research. The back-ground, already quite dense, has been retained as a separate section as it puts the proposed innovation into the context of previous work.

RC: Introduction: I miss a key map, and an image over the study area to get an im-pression how it looks like there.

AR: Both referees asked for an additional image while, at the same time, suggesting shortening the manuscript. As a balance, the revised submission now has an additional figure that shows the character of two field sites (the third being similar to one shown) with overview images and a close-up on the tilt arm showing ground cover. The pre-vious Figure 1 (now Figure 2) has, correspondingly, been shortened by removing the

first panel. Because the geographic arrangement is inconsequential and coordinates are listed, no additional map has been included.

RC: Material and Methods: Now, the tilt arm is explained and introduced as a new field device, and the error range is presented well. See comment above. L. 113 mentions a drill and a references, better to explain the drill if it is important.

AR: The sentence specifying the type of drill has been removed.

RC: Delete the 4.2. paragraph (just one sentence) or incorporate elsewhere. 4.8. the same, paragraph is not necessary.

AR: incorporated 4.2 into preceding paragraph and deleted 4.8

RC: Fig. 4: What is "density" here?

AR: Density is better explained now in Figure 4 and also, values have been removed from the vertical axis as they do not add relevant information to the interpretation.

RC: Acknowledgements: l. 315: "And who came up with those site names..". What is this?

AR: deleted

---

## Author Comment (AC2) · 19 Mar 2020

Thank you very much for your review and your constructive comments on this manuscript. I hope that the explanation given below, and the changes to the manuscript, will provide an adequate response.

Referee comments are indicated as "RC" and author responses as "AR".

RC: I encourage the author to make a short communication exclusively on this topic.

[Figure]

AR: Both referees allude to the possibility of making this a short communication and I respectfully disagree with this suggestion: First, the journal guidelines for (2–4 pages, up to 3 figures/tables, up to 20 references) would not permit an adequate level detail. Second, the potential of processes other than phase change in soil to obfuscate analyses of surface movement can be clearly shown, even with this short dataset and in the absence of additional observations of hydrology.

RC: One of the main unclear points of the manuscript is Figure 4 and the accompanying text, that must be rewritten. In particular, what does 'density' refer to?

AR: Density is better explained now in Figure 4 and also, values have been removed from the vertical axis as they do not add relevant information to the interpretation.

RC: Section 3: Is it possible to add a picture and/or a satellite image of the studied sites?

AR: Both referees asked for an additional image while, at the same time, suggesting shortening the manuscript. As a balance, the revised submission now has an additional figure that shows the character of two field sites (the third being similar to one shown) with overview images and a close-up on the tilt arm showing ground cover. The previous Figure 1 (now Figure 2) has, correspondingly, been shortened by removing the first panel. Because the geographic arrangement is inconsequential and coordinates are listed, no additional map has been included.

RC: Section 4.2 (auxiliary data): Splitting soil temperature data section into a whole paragraph does not seem necessary. Could it be merged into section 4.1?

AR: done

RC: Section 4.8: paragraph not necessary. Could be removed, or used as a small introductory sentence for section 5

AR: removed

RC: Section 5.3: How are defined the thawing and freezing seasons? Soil temperatures, air temperatures, freezing indexes? Moreover, some mathematical notations are unclear, and a few sentences need clarification.

AR: This is explained in the last paragraph of Section 4.7 and the use of soil temperature has now additionally been clarified.

RC: Line122: 'at a distance of 10m or less': I guess it is at a distance of the tilt-arm logger? Please precise.

AR: yes, clarification has been added

RC: Line 172: The wording 'during the time interval [t=0,t]' is unclear (and mathematically awkward) : does z(t) simply denotes the thickness of soil thawed at the time t ? If so, please remove the sentence 'during the time interval [t=0,t]'.

AR: yes, clarified by removing t

RC: Line 178: Should not it be 'proportional' instead of 'equal' in z = sqrt (I(t))?

AR: yes, corrected

RC: Line 195: extra parenthesis in the text (Fig.5)

AR: fixed